# Weak-shot Semantic Segmentation
# via Dual Similarity Transfer

**Junjie Chen**[1], **Li Niu**[1*], **Siyuan Zhou**[1], **Jianlou Si**[2], **Chen Qian**[2], **Liqing Zhang**[1*]

[1]The MoE Key Lab of AI, CSE department, Shanghai Jiao Tong University
[2]SenseTime Research, SenseTime
{chen.bys, ustcnewly, ssluvble}@sjtu.edu.cn
{sijianlou,qianchen}@sensetime.com, zhang-lq@cs.sjtu.edu.cn

## Abstract

Semantic segmentation is an important and prevalent task, but severely suffers from the high cost of pixel-level annotations when extending to more classes in wider applications. To this end, we focus on the problem named weak-shot semantic segmentation, where the novel classes are learnt from cheaper image-level labels with the support of base classes having off-the-shelf pixel-level labels. To tackle this problem, we propose SimFormer, which performs dual similarity transfer upon MaskFormer. Specifically, MaskFormer disentangles the semantic segmentation task into two sub-tasks: proposal classification and proposal segmentation for each proposal. Proposal segmentation allows proposal-pixel similarity transfer from base classes to novel classes, which enables the mask learning of novel classes. We also learn pixel-pixel similarity from base classes and distill such class-agnostic semantic similarity to the semantic masks of novel classes, which regularizes the segmentation model with pixel-level semantic relationship across images. In addition, we propose a complementary loss to facilitate the learning of novel classes. Comprehensive experiments on the challenging COCO-Stuff-10K and ADE20K datasets demonstrate the effectiveness of our method. Codes are available at https://github.com/bcmi/SimFormer-Weak-Shot-Semantic-Segmentation.

## 1 Introduction

Semantic segmentation [27, 6, 45, 47] is a fundamental and active task in computer vision, which aims to produce class label for each pixel in image. Training modern semantic segmentation models usually requires pixel-level labels for each class in each image, and thus costly datasets have been constructed for learning. As a practical task, there is a growing requirement to segment more classes in wider applications. However, annotating dense pixel-level mask is too expensive to cover the continuously increasing classes, which dramatically limits the applications of semantic segmentation.

In practice, we have some already annotated semantic segmentation datasets, *e.g.*, COCO-80 [25]. These costly datasets focus on certain classes (*e.g.*, the 80 classes in [25]), and ignore other uninterested classes (*e.g.*, other classes beyond the 80 classes in [25]). Over time, there may be demand to segment more classes, *e.g.*, extending to COCO-171 [3]. We refer to the off-the-shelf classes having already annotated masks as base classes, and those newly covered classes as novel classes. As illustrated in Fig. 1 (a), one sample in COCO-80 focuses on segmenting *cat*, *bed*, etc, but ignores the *lamp*. Existing scenario for expansion in [3] is to annotate the dense pixel-level masks for novel classes again (*i.e.*, annotating the mask for *lamp* in Fig. 1 (a)), which is too expensive to scale up. To this end, we focus on a cheaper yet effective scenario in this paper, where we only need to annotate image-level labels for the novel classes.

---

*Corresponding author

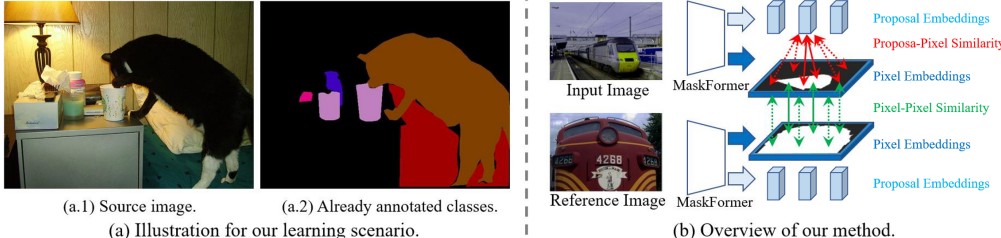

| (a.1) Source image. | (a.2) Already annotated classes. | |
|---|---|---|
| (a) Illustration for our learning scenario. | | (b) Overview of our method. |

Figure 1: In (a), off-the-shelf datasets focus on certain base classes, *i.e.*, the colored masks in (a.2), while leaving novel classes ignored, *i.e.*, the regions in black. We would like to segment novel classes by virtue of cheaper image-level labels. In (b), our method obtains proposal embeddings and pixel embeddings for each image based on MaskFormer [8]. Then, the transferred proposal-pixel similarity could produce masks for proposal embeddings of novel classes, and the transferred pixel-pixel similarity could provide semantic relation regularization for pixel pairs across images. The solid (*resp.*, dotted) arrow indicates similarity (*resp.*, dissimilarity). Best viewed in colors.

We refer to our learning scenario as weak-shot semantic segmentation, which focuses on further segmenting novel classes by virtue of cheaper image-level labels with the support of base classes having pixel-level masks. Specifically, given a standard semantic segmentation dataset annotated only for base classes (the novel classes hide in the ignored regions), we assume that the image-level labels are available for novel classes in each image. We would like to segment all the base classes and novel classes in the test stage. Similar scenario has been explored in RETAB [51], but they 1) further assume that the off-the-shelf dataset contains no novel classes; 2) assume that the *background* class (in PASCAL VOC dataset [10]) consisting of all ignored classes is annotated with pixel-level mask. However, these two assumptions are hard to satisfy in real-world applications, because the ignored region in off-the-shelf dataset may contain novel classes and we actually cannot have the masks of *background* class before having the masks of novel classes. In contrast, our scenario is similar in spirit but more succinct and practical, which is well demonstrated by the aforementioned instance expanding from COCO-80 to COCO-171. Besides, RETAB [51] only conducted experiments on PASCAL VOC, while we focus on more challenging datasets (*i.e.*, COCO-Stuff-10K [3] and ADE20K [50]), without a *background* class annotated with masks.

In weak-shot semantic segmentation, the key problem is how to learn dense pixel-level masks from the image-level labels of novel classes with the support of pixel-level masks of base classes. Our proposed solution is SimFormer, which performs dual similarity transfer upon MaskFormer [8] as shown in Fig. 1 (b). We choose MaskFormer [8] because it disentangles the segmentation task into two sub-tasks: proposal classification and proposal segmentation. Specifically, MaskFormer [8] produces some proposal embeddings (*aka* per-segment embeddings in [8]) from shared query embeddings for input image, each of which is assigned to be responsible for one class present in image (allowing empty assignment). Then MaskFormer performs proposal classification sub-task and proposal segmentation sub-task for each proposal embedding according to their assigned classes.

In our setting and framework, the proposal embeddings of base classes are supervised in both sub-tasks using class and mask annotations, while the proposal embeddings of novel classes are supervised only in classification sub-task due to lacking mask annotations. The proposal segmentation sub-task is essentially learning proposal-pixel similarity. Such similarity belongs to pair-wise semantic similarity, which is class-agnostic and transferable across different categories [7, 36, 5]. Therefore, proposal segmentation for novel classes could be accomplished based on the proposal-pixel similarity transferred from base classes. To further improve the quality of binary masks of novel classes, we additionally propose pixel-pixel similarity transfer, which learns the pixel pair-wise semantic similarity from base classes and distills such class-agnostic similarity into the produced masks of novel classes. In this way, the model is supervised to produce segmentation results containing semantic relationship for novel classes. Inspired by [38, 42, 37], we learn from and distill to cross-image pixel pairs, which could also introduce global context into model, *i.e.*, enhancing the pixel semantic consistency across images. In addition, we propose a complementary loss, based on the insight that the union set of pixels belonging to base classes is complementary to the union set of pixels belonging to novel classes or ignore class in each image, which provides supervision for the union of masks of novel classes.

We conduct extensive experiments on two challenging datasets (*i.e.*, COCO-Stuff-10K [3] and ADE20K [50]) to demonstrate the effectiveness of our method. We summarize our contributions as

1) We propose a dual similarity transfer framework named SimFormer for weak-shot semantic segmentation, in which MaskFormer lays the foundation for proposal-pixel similarity transfer.

2) We propose pixel-pixel similarity transfer, which learns pixel-pixel semantic similarity from base classes and distills such class-agnostic similarity to the segmentation results of novel classes. We also propose a complementary loss to facilitate the mask learning of novel classes.

3) Extensive experiments on the challenging COCO-Stuff-10K [3] and ADE20K [50] datasets demonstrate the practicality of our scenario and the effectiveness of our method.

## 2 Related Works

**Weakly-supervised Semantic Segmentation (WSSS).** Considering the expensive cost for annotating pixel-level masks, WSSS [32, 11, 34, 20] only relies on image-level labels to train the segmentation model, which has attracted increasing attention. The majority of WSSS methods [4, 1, 40] firstly train a classifier to obtain class activation map (CAM) [49] to derive pseudo masks, which are then used to train a standard segmentation model. For example, SEC [18] proposed the principle of "seed, expand, and constrain", which has a great impact on WSSS. Under similar pipeline, some works [22, 4, 40] focus on enhancing the seed, while some other works [17, 39, 46] pay attention to improving the expanding strategy. Although WSSS has achieved great success, the expanded CAM is difficult to cover the intact semantic region due to the lack of informative pixel-level annotations. Fortunately, in our focused problem, such information could be derived from an off-the-shelf dataset and transferred to facilitate the learning of novel classes with only image-level labels.

**Weak-shot Learning.** Reducing the annotation cost is a practical and extensive demand for various applications of deep learning. Recently, weak-shot learning, *i.e.*, learning weakly supervised novel classes with the support of strongly supervised base classes, has been explored in image classification [5], object detection [26, 48, 23], semantic segmentation [51], instance segmentation [16, 19, 2], and so on [35], which has achieved promising success. In weak-shot semantic segmentation [51], the problem is learning to segment novel classes with only image-level labels with the support of base classes having pixel-level labels. Concerning the task setting, as aforementioned, RETAB [51] further assumes that the off-the-shelf dataset has no novel classes and the *background* class is annotated with pixel-level mask, while the setting in this paper is more succinct and practical. Concerning the technical method, RETAB [51] follows the framework of WSSS, which suffers from a complex and tedious multi-stage pipeline, *i.e.*, training classifier, deriving CAM, expanding to pseudo-labels, and re-training. In contrast, we build our framework based on MaskFormer [8] to perform dual similarity transfer, which could achieve satisfactory performance in single-stage without re-training.

**Similarity Transfer.** As an effective method, similarity transfer has been widely applied in various transfer learning tasks [7, 36, 5]. Specifically, semantic similarity (whether the two inputs belong to the same class) is class-agnostic, and thus transferable across classes. To name a few, Guo *et al*. [13] transferred class-class similarity and sample-sample similarity across domains in active learning. CCN [15] proposed to learn semantic similarity between image pair, which is robust in both cross-domain and cross-task transfer learning. PrototypeNet [33] proposed to learn and transfer the similarity between image and prototype across categories for both few-shot classification and zero-shot classification. In this paper, we propose to transfer proposal-pixel similarity and pixel-pixel similarity for weak-shot semantic segmentation. These two types of similarities both belong to semantic similarity, which are highly transferable across classes.

## 3 Methodology

### 3.1 Problem Definition

In our weak-shot semantic segmentation, we are given a standard segmentation dataset annotated for base classes $\mathcal{C}_{\mathrm{b}}$, and we would like to further segment another set of novel classes $\mathcal{C}_{\mathrm{n}}$ ignored in the off-the-shelf dataset, where $\mathcal{C}_{\mathrm{b}} \cap \mathcal{C}_{\mathrm{n}} = \emptyset$. We assume that we have the image-level labels for $\mathcal{C}_{\mathrm{n}}$, which is rather cheaper and more convenient to obtain than pixel-level mask. In summary, for each

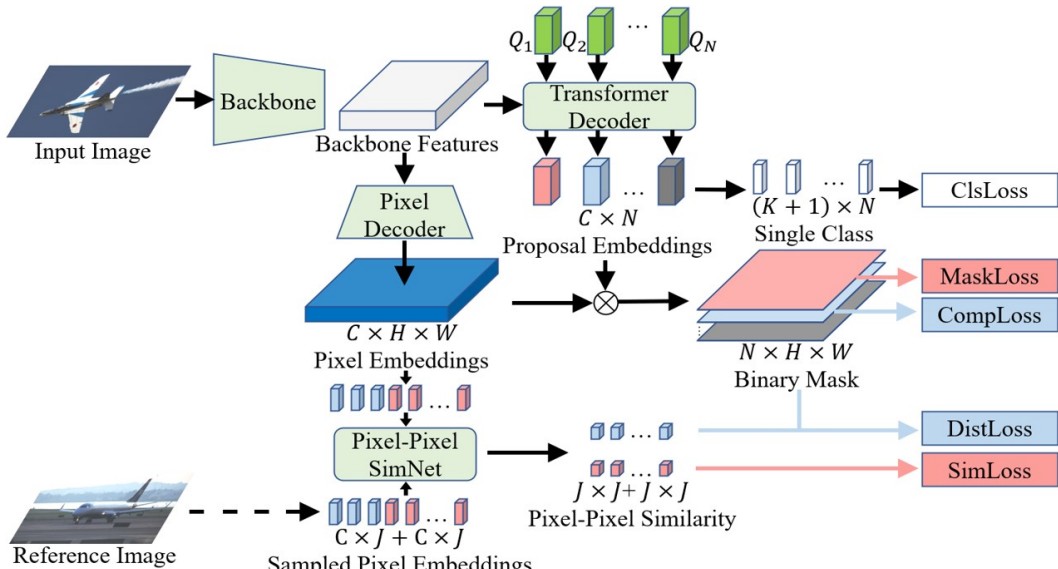

Figure 2: The detailed illustration of our framework. As in MaskFormer, we produce $N$ proposal embeddings in each image. On the one hand, each proposal embedding is fed to the classifier, where both base and novel classes are supervised by classification loss (ClsLoss). On the other hand, the similarities between each proposal embedding and pixel embeddings are computed to produce binary mask, where only base masks (in red) are supervised by GT mask (MaskLoss) while novel masks (in blue) are supervised by complementary loss (CompLoss). We sample some pixels and construct pixel pairs across two images. The concatenated pixel embeddings are fed to SimNet, where the base pixel pairs (in red) are used to train SimNet with similarity loss (SimLoss) and novel pixel pairs (in blue) are used for similarity distillation (DistLoss).

training image, we have image-level labels for both $\mathcal{C}_\mathrm{b}$ and $\mathcal{C}_\mathrm{n}$, and we have pixel-level masks only for $\mathcal{C}_\mathrm{b}$. In the test stage, we need to predict pixel-level masks for both $\mathcal{C}_\mathrm{b}$ and $\mathcal{C}_\mathrm{n}$.

### 3.2 Review of MaskFormer

In this section, we briefly review MaskFormer [8], which lays the foundation of our framework. The general pipeline of MaskFormer disentangles semantic segmentation task into two sub-tasks: proposal classification and proposal segmentation. Specifically, MaskFormer maintains $N$ learnable $C$-dim query embeddings $\mathcal{Q} \in \mathbb{R}^{C \times N}$ shared for all images, as shown in the upper part of Fig. 2. When the image is input, the backbone features are extracted via a backbone. Then, $N$ query embeddings attend to backbone features to produce proposal embeddings $\mathcal{E}_\mathrm{prop} \in \mathbb{R}^{C \times N}$ via a transformer decoder. For each proposal embedding, a bipartite matching algorithm assigns a class present in the input image to it, considering the classification loss and mask loss as the cost for assignment. For proposal classification sub-task, each proposal embedding is fed to a simple classifier to yield class probability predictions $\mathcal{Y} \in \mathbb{R}^{(K+1) \times N}$ over $K$ semantic classes and 1 ignore class. For proposal segmentation sub-task, pixel embeddings $\mathcal{E}_\mathrm{pix} \in \mathbb{R}^{C \times H \times W}$ are extracted from backbone features via a pixel decoder. Afterwards, proposal embeddings are processed by several FC layers and their dot-products with pixel embeddings followed by sigmoid are computed to produce binary masks $\mathcal{M} \in \mathbb{R}^{N \times H \times W}$, i.e., $\mathcal{M}[i, h, w] = \mathrm{sigmoid}(\mathcal{E}_\mathrm{prop}[:, i] \cdot \mathcal{E}_\mathrm{pix}[:, h, w])$. In the training stage, the two sub-tasks for each proposal embedding are supervised by the label and mask of assigned class (the mask loss of ignore class is eliminated). In the test stage, the semantic segmentation result for each class at pixel $(h, w)$ is obtained by summarizing all the masks weighted by the class score, i.e., $\arg\max_{c \in \{1,...,K\}} \sum_{i=1}^{N} \mathcal{Y}[c, i] \cdot \mathcal{M}[i, h, w]$. For more details, please refer to MaskFormer [8].

### 3.3 Proposal-Pixel Similarity Transfer on MaskFormer

In our setting, we have only image-level labels for novel classes and we choose to produce pixel-level masks via proposal-pixel similarity transfer based on MaskFormer [8]. For concise description, we

refer to the proposal embeddings assigned with base (*resp.*, novel or ignore) classes as base (*resp.*, novel or ignore) proposal embeddings. We refer to the binary masks produced from base (*resp.*, novel or ignore) proposal embeddings as base (*resp.*, novel or ignore) masks. We refer to the image pixels actually belonging to base (*resp.*, novel or ignore) classes as base (*resp.*, novel or ignore) pixels.

Our method makes full use of the disentangled two sub-tasks of MaskFormer to perform proposal-pixel similarity transfer. The proposal classification sub-task assigns class labels to proposal embeddings. The proposal segmentation sub-task calculates the similarity between each proposal embedding and all pixels to obtain a similarity map, which is the segmentation mask corresponding to the class of this proposal embedding. Such proposal-pixel similarity is class-agnostic and thus transferable across classes. In this way, our method could produce novel masks via the similarity transferred from base classes.

Our framework is illustrated in Fig. 2, where the upper part is the same as MaskFormer producing proposal embeddings $\mathcal{E}_{prop} \in \mathbb{R}^{C \times N}$. Because we have no mask for novel classes in the bipartite matching algorithm, we omit the mask loss in the cost if the class to be assigned belongs to $\mathcal{C}_\mathrm{n}$. For the proposal classification sub-task, all proposal embeddings are supervised to predict the assigned classes because we have all the image-level labels of each input image. Thus, we have the classification loss

$$\mathcal{L}_\mathrm{cls} = \sum_{i=1}^{N} -\log(\mathcal{Y}[y^*(i), i]), \tag{1}$$

where $y^*(i)$ indicates the assigned single ground-truth (GT) class of the $i$-th proposal embedding. For the binary segmentation sub-task, only the base masks are supervised as

$$\mathcal{L}_\mathrm{mask} = \sum_{y^*(i) \in \mathcal{C}_\mathrm{b}} \mathcal{L}_\mathrm{focal+dice}(\mathcal{M}[i, :, :], M^*(y^*(i))), \tag{2}$$

where $M^*(y^*(i))$ indicates the GT mask of the assigned class of the $i$-th proposal, and $\mathcal{L}_\mathrm{focal+dice}$ is a binary mask loss consisting of a focal loss [24] and a dice loss [29] as in MaskFormer [8]. Under the supervision of mask loss, the base proposal embeddings will produce binary similarities with all the pixel embeddings, *i.e.*, each proposal embedding is *similar* or *dissimilar* to each pixel embedding. Such semantic pair-wise similarity is class-agnostic, and thus transferable across classes. Therefore, although the novel proposal embeddings are not supervised by GT masks, we can calculate the similarity between novel proposal embeddings and pixel embeddings to produce novel masks.

### 3.4 Pixel-Pixel Similarity Transfer via Pixel SimNet

Due to lacking semantic supervision of novel classes, the novel masks are hard to precisely reach semantic consistency across training images, so we design a novel strategy to learn the pair-wise semantic similarity from base pixels across images and distill such class-agnostic similarity to novel pixels across images, which could complement the above proposal-pixel similarity.

Specifically, to learn pixel-pixel similarity across images, for each input image, we randomly sample another training image as its reference image. According to image-level labels, we ensure that the input image and reference image have common base and novel classes. We train a pixel-pixel similarity net (SimNet) using base pixels which are associated with GT pixel-level labels. By class-aware random sampling $J$ base pixels from both images, we construct $J \times J$ ($J = 100$ in our experiments) pixel pairs balanced in binary semantic class (*i.e.*, *similar v.s. dissimilar*). We concatenate the pixel embeddings of paired base pixels to obtain pixel-pair embeddings $\mathcal{E}_\mathrm{pair-b} \in \mathbb{R}^{(C+C) \times J \times J}$. A SimNet (6 FC layers followed by sigmoid function) takes in $\mathcal{E}_\mathrm{pair-b}$ and outputs the similar scores $\mathcal{R}_\mathrm{b} \in \mathbb{R}^{J \times J}$ for the base pixel pairs. We have GT labels for the pixels from base region, and thus $\mathcal{R}_\mathrm{b}$ is supervised by simple binary classification loss $\mathcal{L}_\mathrm{sim}$ for training SimNet.

To distill the learned pixel-pixel semantic similarity into the segmentation of novel pixels, we also construct $J \times J$ pixel pairs from the not-base region (ignore pixels are much fewer than novel pixels) of both images respectively and obtain $\mathcal{E}_\mathrm{pair-n} \in \mathbb{R}^{(C+C) \times J \times J}$. Because the input image and reference image have common novel classes, the sampled pixel pairs have chance to belong to the same novel classes, which could inject semantic consistency (*i.e.*, *similar* class) into novel pixels. We use the SimNet to estimate the GT semantic similarity and obtain $\mathcal{R}_\mathrm{n} \in \mathbb{R}^{J \times J}$, which is detached of gradient and employed as the distillation source. For the distillation target, we compute

the segmentation scores of novel classes, *i.e.*, $\mathcal{S}_{\mathrm{n}} = \{\sum_{i=1}^{N} \mathcal{Y}[c,i] \cdot \mathcal{M}[i,:,:]\}_{c \in \mathcal{C}_{\mathrm{n}}}$, and collect the segmentation scores of sampled pixels in the input image $\mathcal{S}_{\mathrm{input-n}} \in \mathbb{R}^{|\mathcal{C}_{\mathrm{n}}| \times J}$ and those in the reference image $\mathcal{S}_{\mathrm{ref-n}} \in \mathbb{R}^{|\mathcal{C}_{\mathrm{n}}| \times J}$. The pixel-pixel similarity distillation from base classes to novel classes is achieved by

$$\mathcal{L}_{\mathrm{dist}} = \mathrm{BCE}(\mathrm{ReLU}(\mathrm{COS}_{\mathrm{enum}}(\mathcal{S}_{\mathrm{input-n}}, \mathcal{S}_{\mathrm{ref-n}})), \mathcal{R}_{\mathrm{n}}), \tag{3}$$

where $\mathrm{COS}_{\mathrm{enum}}(\cdot, \cdot)$ computes the cosine similarity of all enumerated pixel pairs between the two sets of $|\mathcal{C}_{\mathrm{n}}|$-dim vectors. $\mathrm{ReLU}(\cdot)$ focuses on the non-negative values, since zero value means that two pixels are independent and thus dissimilar. $\mathrm{BCE}(\cdot, \cdot)$ is the binary cross-entropy loss to push the distillation target towards the distillation source. In this way, the pixel-pixel similarity is learned from base classes and transferred to facilitate the semantic consistency of novel masks across images.

### 3.5 Complementary Loss

Although we have no pixel-level annotation for novel classes, we have the prior knowledge that the union set of novel/ignore pixels and the union set of base pixels are complementary in each image. Therefore, we use a complementary loss to inject such insight into our model:

$$\mathcal{L}_{\mathrm{comp}} = \mathcal{L}_{\mathrm{focal+dice}}(\mathbf{1} - \bigcup_{y^*(i) \in \mathcal{C}_{\mathrm{b}}} M^*(y^*(i)), \bigcup_{y^*(i) \in \mathcal{C}_{\mathrm{n}}+\{\mathrm{ignore}\}} \mathcal{M}[i,:,:]), \tag{4}$$

where the first input is the complementary set of the union set of all the GT masks of base classes, the second input is the union set of all the binary masks produced by the proposal embeddings assigned with novel classes or ignore class, and the union operator specifically takes the maximum value over involved binary masks at each pixel. Because the ignore mask produced from ignore proposal embedding is meaningless here as in MaskFormer [8], we reset all the values in the ignore mask as a constant value $\gamma$ (0.1 by default), which is a prior probability for the ignore class. The constant prior probability is similar to the background threshold used in weakly supervised semantic segmentation [21, 40, 43].

### 3.6 Training and Inference

Our training and inference pipelines are the same as MaskFormer [8], and the full training loss is

$$\mathcal{L}_{\mathrm{full}} = \mathcal{L}_{\mathrm{cls}} + \mathcal{L}_{\mathrm{mask}} + \mathcal{L}_{\mathrm{sim}} + \alpha \mathcal{L}_{\mathrm{dist}} + \beta \mathcal{L}_{\mathrm{comp}}, \tag{5}$$

where $\mathcal{L}_{\mathrm{cls}}$ and $\mathcal{L}_{\mathrm{mask}}$ implicitly include the default trade-off hyper-parameters in MaskFormer, and $\mathcal{L}_{\mathrm{sim}}$ is actually also a classification loss and balances well with $\mathcal{L}_{\mathrm{cls}}$. We use $\alpha$ (0.1 by default) for balancing the distillation loss and $\beta$ (0.2 by default) for balancing the complementary loss.

## 4 Experiments

### 4.1 Datasets, Evaluation, and Implementation Details

We explore the learning scenario and evaluate our method on two challenging datasets. COCO-Stuff-10K [3] contains 9k training images and 1k test images, covering 171 semantic classes. ADE20K [50] has 20k training images and 2k validating images, covering 150 semantic classes. All the data preparing processes follow those in MaskFormer [8]. For the learning scenario, we random split all the classes in the dataset into base classes and novel classes. Then we keep the image-level labels of novel classes in each training image, and reset the pixel-level masks of novel classes to ignore class (*e.g.*, class ID 255). We follow the split ratio $|\mathcal{C}_{\mathrm{b}}| : |\mathcal{C}_{\mathrm{n}}| = 3 : 1$ in RETAB [51], which is also widely used in few-/zero-shot learning [31, 12]. We employ Intersection-over-Union (IoU) as our evaluation metric. To show the performance more clearly, we focus on mean IoU of novel classes, because the base classes are always supervised by GT masks and exhibit robust performances. For the network architecture, we follow the setting of MaskFormer and use ResNet50 [14] pre-trained on ImageNet [9] as backbone. More details are left to Appendix A1, A2, and A3.

### 4.2 Comparison with Prior Works

**Comparative Baselines.** We set three groups of baselines: 1) representative WSSS methods, including SEAM [40], CONTA [43], CPN [44], and RIB [21]; 2) weak-shot segmentation segmentation

Table 1: Comparison of various methods on two benchmark datasets in four splits. The methods marked by * are appended with a re-training stage. The best results are highlighted in boldface.

| Method | COCO-Stuff-10K [3] | | | | ADE20K [50] | | | |
|---|---|---|---|---|---|---|---|---|
| | Split1 | Split2 | Split3 | Split4 | Split1 | Split2 | Split3 | Split4 |
| SEAM* [40] | 20.4 | 16.6 | 22.3 | 17.7 | 26.4 | 22.7 | 19.8 | 20.7 |
| CONTA* [43] | 21.7 | 17.2 | 23.1 | 18.7 | 27.5 | 24.1 | 20.4 | 21.2 |
| CPN* [44] | 21.6 | 18.4 | 23.7 | 19.2 | 27.9 | 24.7 | 20.6 | 21.5 |
| RIB* [21] | 22.3 | 18.2 | 24.3 | 19.5 | 29.5 | 26.4 | 21.1 | 21.8 |
| RETAB* [51] | 27.5 | 23.1 | 29.8 | 25.6 | 34.6 | 31.3 | 25.7 | 25.9 |
| SimFormer | 31.1 | 26.0 | 31.5 | 29.5 | 36.4 | 33.4 | 28.2 | 28.5 |
| SimFormer* | **33.5** | **27.4** | **33.7** | **31.7** | **37.6** | **35.3** | **29.5** | **31.8** |
| FullyOracle | 37.5 | 34.0 | 43.8 | 36.0 | 47.8 | 44.3 | 38.3 | 36.2 |

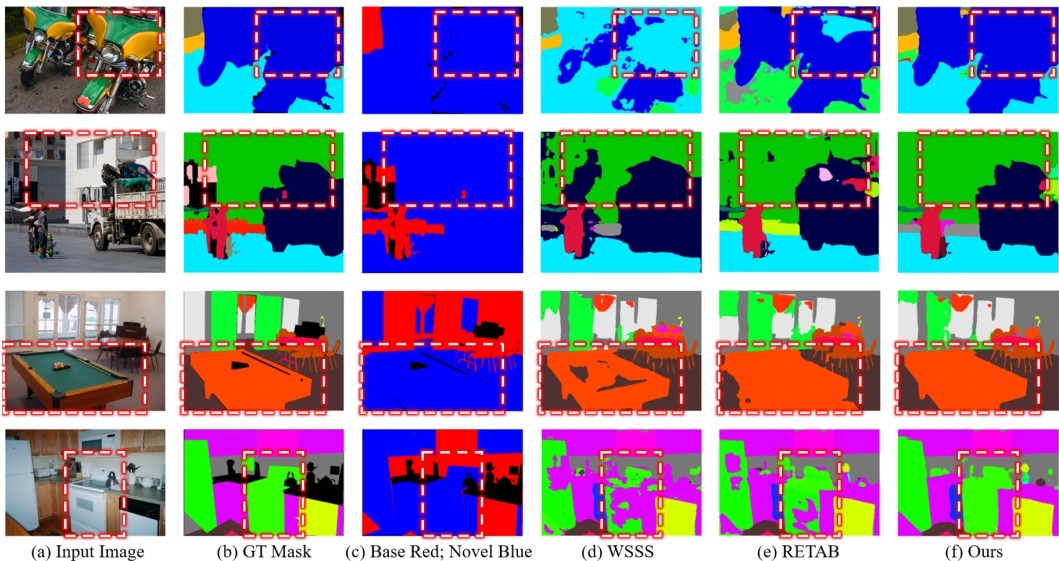

Figure 3: Visual comparison on COCO-Stuff-10K (the first two rows) and ADE20K (the last two rows) datasets. Columns (a,b) show input images and GT masks. Column (c) shows the base/novel split maps indicating base pixels by red and novel pixels by blue. Columns (d,e,f) show the results of three methods. The dotted rectangles highlight the major differences.

methods, including RETAB [51]; 3) the standard segmentation method, a MaskFormer [8] trained with GT masks of novel classes, named *FullyOracle*. Because WSSS baselines and RETAB all need a re-training stage to achieve the final performance and there are GT masks of base classes in training images, we re-train all baselines with mixed labels. Specifically, the mixed labels consist of GT masks of base classes and pseudo-labels of novel classes generated by the model (*e.g.*, deriving and expanding CAM). For a fair comparison, we re-train each baseline on the standard MaskFormer [8] using the generated mixed labels. Therefore, all the methods are consistent in network architecture. Our method could also be appended with a re-training stage for further improvement. We use the mixed labels consisting of GT masks of base classes and pseudo-labels of novel classes predicted by our model (instead of using CAM) in the re-training stage.

**Quantitative Comparison.** We summarize all the results on four random splits of two datasets in Tab. 1. Even without re-training, our method can dramatically outperform WSSS baselines and RETAB [51] (*e.g.*, about 2.7% against RETAB), demonstrating the effectiveness of our method based on dual similarity transfer. By appending a re-training stage as in WSSS, our method is further improved by about 1.9%. Compared with the fully-supervised upper bound *Fully Oracle*, our method with re-training could achieve satisfactory performances (*i.e.*, about $77\% \sim 89\%$ of the upper bound)

Table 2: Module contributions on four splits of COCO-Stuff-10K. "Pr" is proposal-similarity transfer, "Pi" is pixel-pixel similarity transfer, and "Co" is complementary loss.

| Pr | Pi | Co | S1 | S2 | S3 | S4 |
|----|----|----|------|------|------|------|
| √ |   |   | 23.5 | 19.4 | 26.6 | 22.0 |
| √ | √ |   | 27.4 | 23.7 | 29.4 | 26.6 |
| √ |   | √ | 27.0 | 23.6 | 29.1 | 25.6 |
| √ | √ | √ | 31.1 | 26.0 | 31.5 | 29.5 |

Table 3: The transferability of pixel-pixel similarity on four splits of COCO-Stuff-10K. We use F1-score as the metric for the binary classification task (*i.e.*, *Dissimilar v.s. Similar*).

| | | S1 | S2 | S3 | S3 |
|-------|------|------|------|------|------|
| Base | *Dis* | 86.7 | 87.8 | 93.9 | 94.7 |
| | *Sim* | 89.3 | 89.1 | 86.6 | 87.4 |
| Novel | *Dis* | 80.4 | 79.5 | 80.5 | 83.7 |
| | *Sim* | 75.6 | 73.1 | 73.4 | 78.5 |

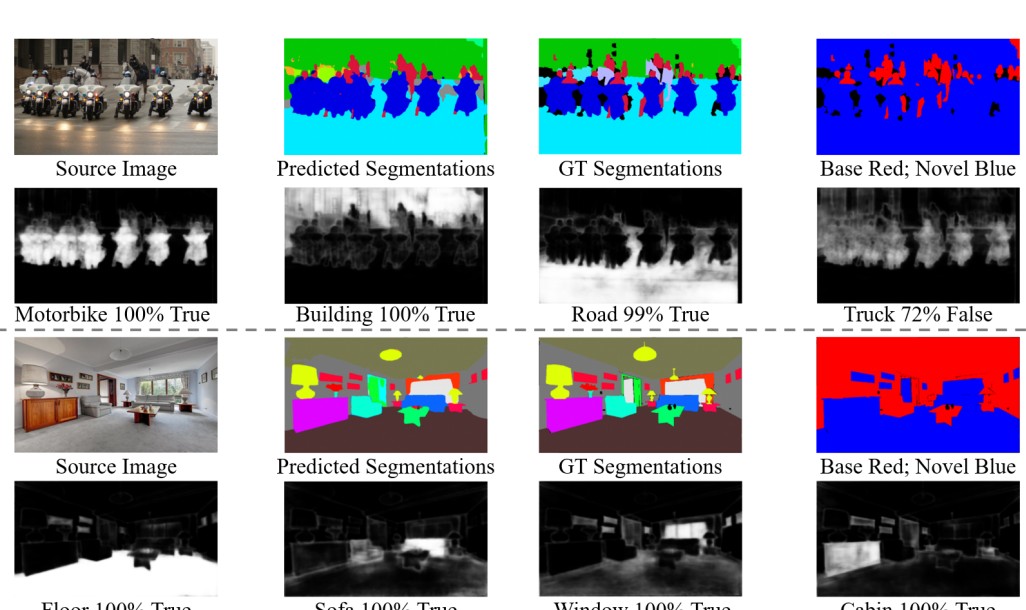

Figure 4: Visualizations for the novel proposals on COCO-Stuff-10K (upper) and ADE20K (lower) datasets. In each example, the first row shows the source image, predicted segmentation, GT segmentation, and base/novel split map, and the second row shows the first four proposals ranked by the predicted scores of novel classes. The caption of each proposal sub-figure describes: 1) the class name; 2) the predicted class score; 3) whether the class is actually present in the image.

on the two challenging datasets, showing the potential of our dual similarity transfer framework for learning novel classes from only image-level labels.

**Qualitative Comparison.** We conduct qualitative comparison for the re-trained WSSS baseline RIB [21], RETAB [51], and our method on both datasets. As shown in Fig. 3, WSSS baseline tends to produce incomplete masks, and our method could produce more complete and precise semantic masks against RETAB [51]. Even in complex scenes (*e.g.*, the 2nd and 3rd rows), our method still predicts preferable results for novel classes. More visualizations are left to Appendix A4.

### 4.3 Module Analysis

**Proposal-pixel Similarity Transfer.** As shown in the 1st row of Tab. 2, only with proposal-pixel similarity transfer, the model could achieve acceptable performance on various dataset splits, which lays the foundation of our method. We provide in-depth qualitative visualization in Fig. 4, from which we could directly inspect the single-label classification and binary segmentation sub-tasks of each proposal embedding. Overall, the predicted classes are precise and confident, and the produced masks of proposal embeddings completely cover the corresponding semantic classes. Although *Truck* is actually not in the first example, the class score and binary mask are both relatively lower, and thus the fused result will not severely degrade the final segmentation performance.

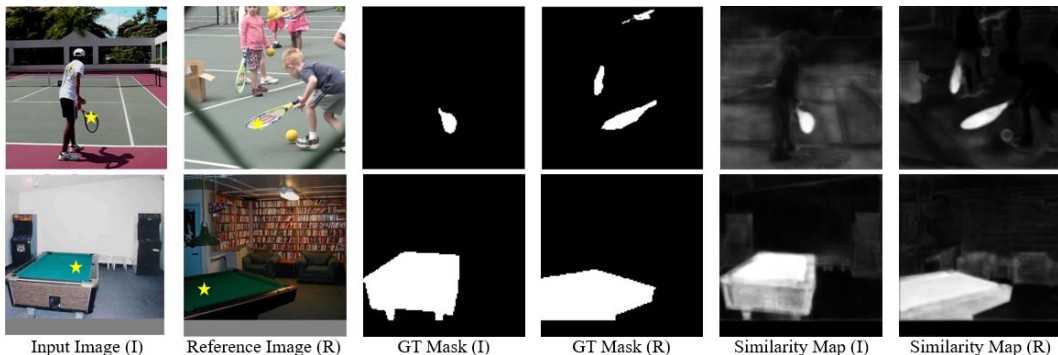

| Input Image (I) | Reference Image (R) | GT Mask (I) | GT Mask (R) | Similarity Map (I) | Similarity Map (R) |

Figure 5: Visualizations for the pixel-pixel similarity on two datasets. In each row, the left two columns show the input image (I) and the reference image (R), where the yellow stars indicate the sampled pixels. The middle two columns depict the GT masks of the corresponding novel class. In the right two columns, we use SimNet to estimate the similarities between the sampled pixel in I (*resp.*, R) and all pixels in R (*resp.*, I), and get the similarity map (R) (*resp.*, (I)).

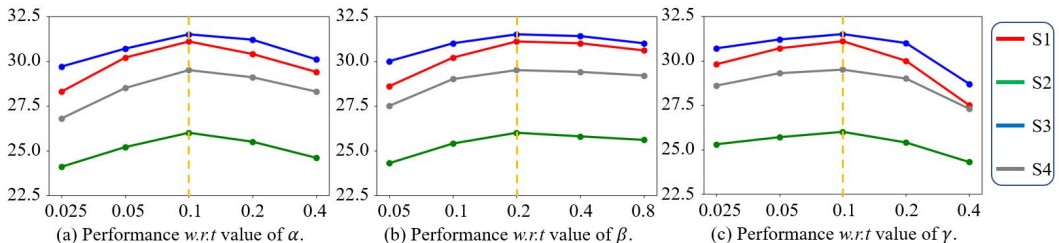

(a) Performance *w.r.t* value of $\alpha$.     (b) Performance *w.r.t* value of $\beta$.     (c) Performance *w.r.t* value of $\gamma$.

Figure 6: The performances of our model *w.r.t* different values of $\alpha$, $\beta$, and $\gamma$ on four splits of COCO-Stuff-10K dataset. The dotted lines indicate the default values.

**Pixel-pixel Similarity Transfer.** As shown in the 2nd row of Tab. 2, we solely enable pixel-pixel similarity transfer on the foundation, which leads to dramatical improvement. The in-depth quantitative analysis about the transferability is shown in Tab. 3. We collect base pixel pairs and novel pixel pairs sampled from 100 training image pairs, and employ GT labels to evaluate the similarities predicted by our SimNet. From the table, we can find that the semantic similarity learned from base classes is highly transferrable across classes. The in-depth qualitative visualization is given in Fig. 5, where we can see that for each sampled pixel, SimNet could "activate" the pixels belonging to the same semantic class across images. Therefore, the transferred pixel-pixel similarity could provide effective semantic relationship for distilling to regularize novel classes. The Fig. 6 (a) shows the impact of $\alpha$ (controlling the strength of distillation) to our model. Specifically, we change $\alpha$ in a range while fixing the other hyper-parameters, and depict the results obtained by our full-fledged model. Thus, our method is relatively robust to $\alpha$ in an appropriate range. Other experimental analyses including reference image selection and pixel number selection are left to Appendix A5.

**Complementary Loss.** As shown in the 3rd row of Tab. 2, we solely enable our complementary loss on the foundation, which also results in obvious promotion. In the 4th row of Tab. 2, with all the modules enabled, our full-fledged model achieves the optimal performance. The Fig. 6 (b,c) illustrate the impact of $\beta$ (controlling the strength of injecting "complementary" insight) and $\gamma$ (the prior probability for the ignore class) to our model. Larger $\gamma$ will block the supervision to novel masks and degrade the performance. Smaller $\gamma$ will "over-activate" novel classes on ignore pixels, which has relatively less impact. Overall, our model is robust to $\beta$ and $\gamma$ in an appropriate range.

## 4.4 Generalization Ability for Dataset Expansion

Focusing on a problem that uses cheaper labels to expand the off-the-shelf dataset, we naturally wonder the generalization ability of our method in such expansion. We explore by further manipulating the datasets, *e.g.*, considering more novel classes. The experiments are left to Appendix A6.

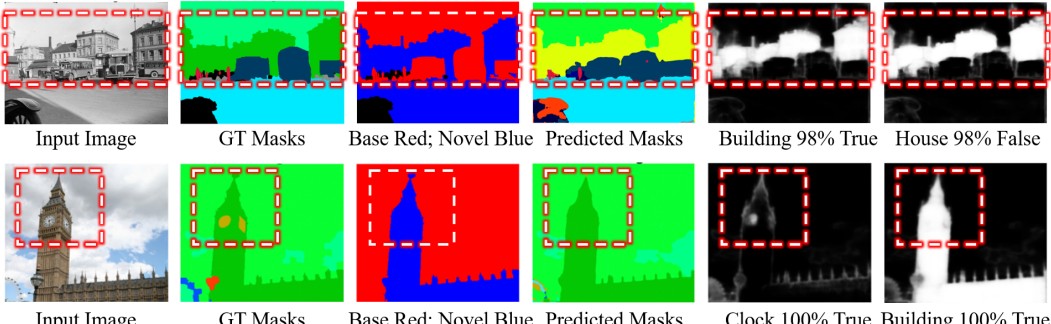

| Input Image | GT Masks | Base Red; Novel Blue | Predicted Masks | Building 98% True | House 98% False |

| Input Image | GT Masks | Base Red; Novel Blue | Predicted Masks | Clock 100% True | Building 100% True |

Figure 7: Illustration for two failure cases. The left two columns depict the input images and GT masks. The middle two columns indicate the base/novel split maps and the semantic masks predicted by our model. The right two columns show the in-depth two sub-tasks of the proposals which mainly lead to the corresponding failure.

### 4.5 Limitation Discussion and Failure Case

Weak-shot semantic segmentation is prone to suffer from various issues due to lacking strong supervision for novel classes. Our method is built based on MaskFormer to disentangle the segmentation task into two sub-tasks and depends on similarity transfer for learning novel classes. Therefore, we summarize two major problems found in practice.

Firstly, the model may mistakenly classify the proposal, as illustrated in the 1st row of Fig. 7. Probably because "building" and "house" are two fine-grained novel classes, the model predict both classes with high confidences. Unfortunately, the "house" outperforms slightly, but which actually does not exist in the image. Therefore, even the proposal segmentation sub-task is done well, the final segmentation result is misled by the proposal classification sub-task.

Secondly, the model may fail in proposal segmentation sub-task, as shown in the 2nd row of Fig. 7. The proposal classification sub-task is done well for both "clock" and "building" classes. Actually, the model has properly predicted the binary mask of "clock". However, probably because the "clock" is too small in image, the binary mask of "clock" is too weak compared with "building". Thus, the "clock" is covered by "building" in the final segmentation result.

Anyway, such limitation and problem exist mainly due to lacking strong and accurate labels for novel classes. In such challenging cases, similarity transfer may be not effective enough for producing perfect results. We would like to further alleviate such issues by exploiting more transferrable targets across classes in future works.

## 5 Conclusion

In this paper, we focus on the problem of weak-shot semantic segmentation, which makes full use of the off-the-shelf pixel-level labels of base classes to support segmenting novel classes with image-level labels. For such problem, we have proposed a dual similarity transfer framework, which is built upon MaskFormer for performing proposal-pixel similarity transfer. To further regularize the model with pixel-level semantic relationship, we learn pixel-pixel similarity from base classes and distill such class-invariant similarity to novel masks. We have also proposed a complementary loss for facilitating the mask learning of novel classes. Extensive experiments on the challenging COCO-Stuff-10K and ADE20K datasets have demonstrated the effectiveness of our proposed method.

## Acknowledgements

The work was supported by the National Science Foundation of China (62076162), and the Shanghai Municipal Science and Technology Major/Key Project, China (2021SHZDZX0102, 20511100300).

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
