# Supplementary of Weak-shot Semantic Segmentation via Dual Similarity Transfer

**Junjie Chen[1], Li Niu[1]\*, Siyuan Zhou[1], Jianlou Si[2], Chen Qian[2], Liqing Zhang[1]\***
[1]The MoE Key Lab of AI, CSE department, Shanghai Jiao Tong University
[2]SenseTime Research, SenseTime
{chen.bys, ustcnewly, ssluvble}@sjtu.edu.cn
{sijianlou,qianchen}@sensetime.com, zhang-lq@cs.sjtu.edu.cn

## Appendix

In this appendix, we first clarify more details about the datasets, evaluation, and implementation in Section A1, Section A2, and Section A3. Afterwards, we provide more qualitative comparisons in Section A4. Then, we conduct more experiments about pixel-pixel similarity transfer in Section A5. Finally, we conduct experiments to explore the generalization ability of our model to dataset expansion in Section A6.

## A1. Datasets

For dataset setting, we generally follow MaskFomer [3], and choose two representative and challenging datasets, *i.e.*, COCO-Stuff-10K [1] and ADE20K [13]. These two datasets both contain enough classes and abundant images, which are appropriate for exploring the problem about transfer learning across classes. Specifically, COCO-Stuff-10K [1] totally covers 171 semantic-level classes. The 10k images are split as $9 : 1$ for training and testing respectively. The images of COCO-Stuff-10K [1] dataset come from the original COCO dataset [5]. ADE20K [13] totally contains 150 semantic-level categories, and includes 20k images for training and 2k images for validation. The images belong to the ADE20K-Full dataset where 150 semantic classes are selected to be covered in evaluation from the SceneParse150 challenge.

## A2. Evaluation

We use the standard evaluation API in Detectron2 [10] and MaskFormer [3] for computing Intersection-over-Union (IoU). After obtaining the IoU of all classes, we average the IoU of novel classes as the evaluation metric for final performance. Because base classes always have GT masks for supervision in various methods and we find that the performances of base classes are relatively robust in practice, we focus on the performance of novel classes for evaluation.

## A3. Implementation Details

The training and inference settings generally follow these in Detectron2 [10] and MaskFormer [3] for corresponding dataset. Specifically, we adopt AdamW [6] with the poly [2] learning-rate schedule. We set the initial learning rate as $10^{-4}$ and the weight decay as $10^{-4}$. For data augmentation, we include the random horizontal flipping, random color jittering, random scale jittering (between 0.5 and 2.0), as well as random cropping. For the COCO-Stuff-10k dataset [1], we adopt the crop size $640 \times 640$, the batch size 8 and train all methods for totally 60k iterations. For the ADE20K dataset

---

\*Corresponding author

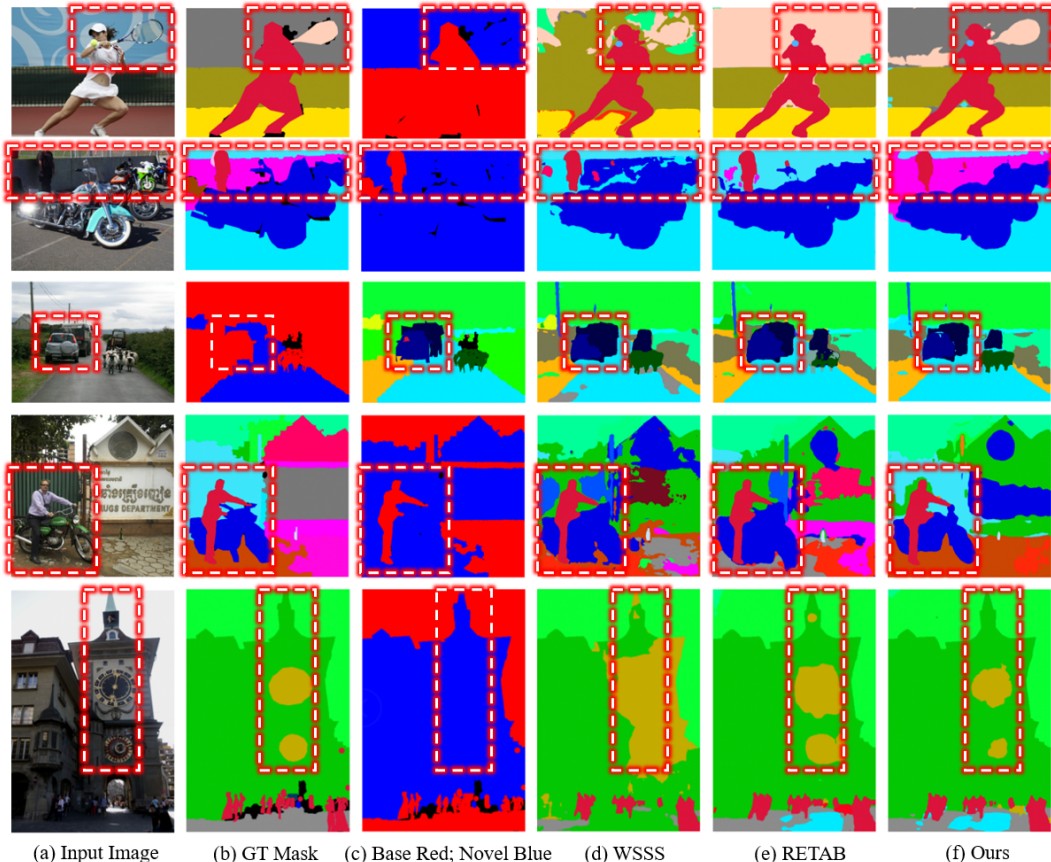

| (a) Input Image | (b) GT Mask | (c) Base Red; Novel Blue | (d) WSSS | (e) RETAB | (f) Ours |

Figure 1: Visual comparison on COCO-Stuff-10K dataset. Columns (a,b) show input images and GT masks. Column (c) shows the base/novel split maps indicating base pixels by red and novel pixels by blue. Columns (d,e,f) show the results of three methods. The dotted rectangles highlight the major differences.

[13], we employ the crop size $512 \times 512$, the batch size $8$ and train all methods for totally 160k iterations. In the test stage, we resize the shorter side of the image to the corresponding crop size. The proposed method is implemented based on the codebase of MaskFormer [3]. Specifically, we use Python $3.7$, PyTorch $1.8.0$ [7], and Detectron2 $0.6$ [10]. For the system environment, we conduct experiments on Ubuntu $18.04$ with 32 GB Intel 9700K CPU and four NVIDIA 3090 GPUs. The random seed is set as $0$ for all experiments if not stated otherwise.

## A4. More Qualitative Comparisons

In this section, we provide more qualitative comparison for the re-trained WSSS baseline RIB [4], RETAB [14] as well as our method on both datasets. As shown in Fig. 1 and Fig. 2, our method could produce superior semantic masks for novel classes against WSSS baseline and RETAB. Specifically, WSSS baseline and RETAB may suffer from "under-expansion" or "over-expansion" problem due to the expansion on noisy CAM [12]. As a consequence, they may produce incomplete semantic masks (*e.g.*, the 2nd row in Fig. 1) or oversize semantic masks (*e.g.*, the 5th row in Fig. 1). In contrast, our method depends on similarity transfer based on MaskFormer [3], which could predict preferable segmentation results. Even in complex scenes having large areas of novel classes (*e.g.*, the 4th row of Fig. 1 and the 4th row of Fig. 2), our method still produces satisfactory results for novel classes, demonstrating the robustness of our method.

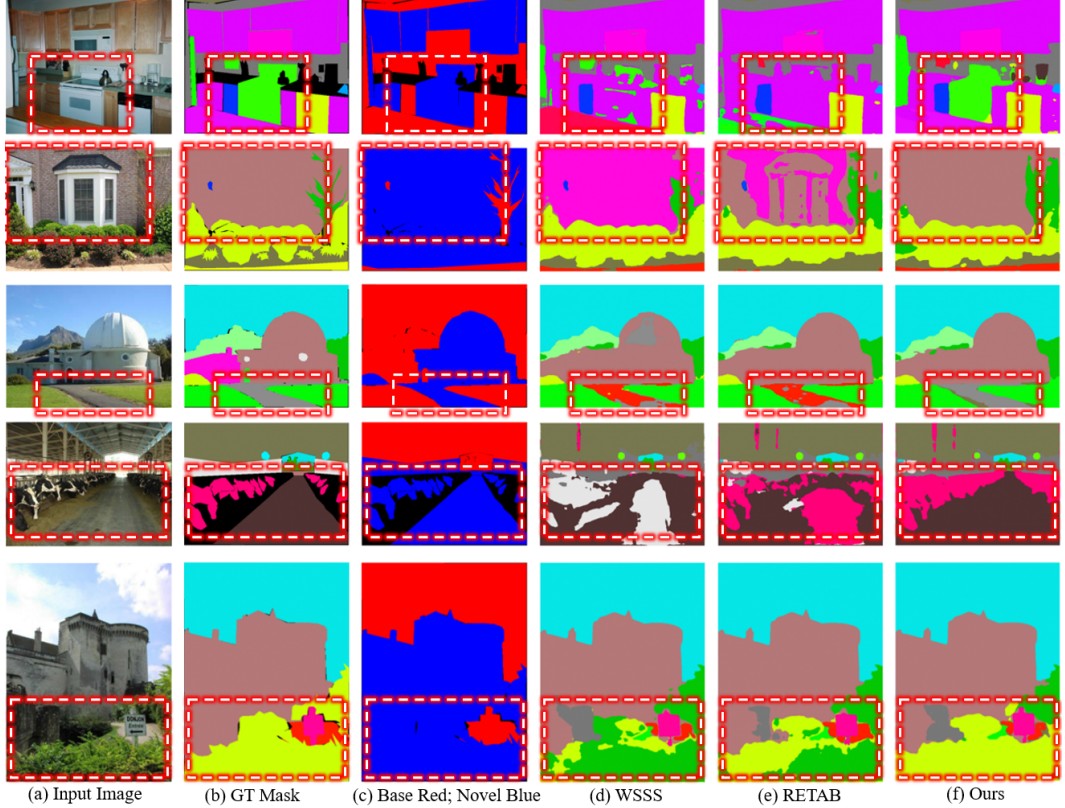

| (a) Input Image | (b) GT Mask | (c) Base Red; Novel Blue | (d) WSSS | (e) RETAB | (f) Ours |

Figure 2: Visual comparison on ADE20K dataset. Columns (a,b) show input images and GT masks. Column (c) shows the base/novel split maps indicating base pixels by red and novel pixels by blue. Columns (d,e,f) show the results of three methods. The dotted rectangles highlight the major differences.

## A5. Additional Experiments about Pixel-Pixel Similarity Transfer

As described in the main paper, we respectively sample $J$ pixels from input image and reference image, and construct $J \times J$ pixel pairs for learning and distilling pixel-pixel similarity. The impact of $J$ is summarized in Tab. 1, where we could see that our model is relatively robust to the value $J$. Larger $J$ leads to more dense pixel pairs, but suffers from larger computation memory and higher complexity in the training stage. Considering that larger $J$ only improves the performance slightly, our default value $J = 100$ is a reasonable choice.

As mentioned in the main paper, we construct pixel pairs across images, which could also introduce a "global context" as discussed in [9, 11, 8]. In Tab. 2, we report a version constructing the pixel pairs within the input image, named "self-pair", *i.e.*, using the input image itself as the reference image. "Cross-Pair" is our full-fledged model using pixel pairs across images, while "Basic Model" is our full-fledged model without pixel-pixel similarity transfer. As shown in Tab. 2, "Self-Pair" only slightly improves the basic model. Therefore, enhancing semantic consistency across images could benefit the model more significantly.

## A6. Generalization Ability to Dataset Expansion

As discussed in the main paper, our weak-shot semantic segmentation focuses on making full use of the off-the-shelf datasets of base classes to support further segmenting novel classes with only cheaper image-level annotations. Therefore, the expansion in class number is a practical scenario in our focused problem. In this section, we additionally include more splits to investigate the generalization ability of our model in scenarios containing more novel classes. As shown in Tab. 3, from Split 5 to Split 9, we progressively move random base classes to novel classes based on the original class split

Table 1: The impact of sample pixel number $J$ on our model on four splits of COCO-Stuff-10K dataset.

| Point Number | S1 | S2 | S3 | S4 |
|:---:|:---:|:---:|:---:|:---:|
| $J = 50$ | 30.7 | 25.6 | 31.1 | 29.2 |
| $J = 100$ | 31.1 | 26.0 | 31.5 | 29.5 |
| $J = 150$ | 31.2 | 26.3 | 31.6 | 29.6 |

Table 2: The performance of different configurations of our model on four splits of COCO-Stuff-10K dataset.

| Configuration | S1 | S2 | S3 | S4 |
|:---:|:---:|:---:|:---:|:---:|
| Basic Model | 27.0 | 23.6 | 29.1 | 25.6 |
| Self-Pair | 27.8 | 24.3 | 30.0 | 26.6 |
| Cross-Pair | 31.1 | 26.0 | 31.5 | 29.5 |

Table 3: Performances of splits containing more novel classes on COCO-Stuff-10K dataset [1]. The base ratio (*resp.*, novel ratio) of each split indicates the ratio of base classes (*resp.*, novel classes) in all 171 classes.

| | Split 4 | Split 5 | Split 6 | Split 7 | Split 8 | Split 9 |
|:---:|:---:|:---:|:---:|:---:|:---:|:---:|
| Base Ratio | 75% | 70% | 65% | 60% | 55% | 50% |
| Novel Ratio | 25% | 30% | 35% | 40% | 45% | 50% |
| RETAB* | 25.6 | 23.0 | 22.5 | 21.5 | 19.9 | 19.8 |
| SimFormer | 29.5 | 27.1 | 26.3 | 25.1 | 23.9 | 23.2 |
| SimFormer* | 31.7 | 28.3 | 28.2 | 25.9 | 24.3 | 24.0 |
| FullyOracle | 36.0 | 35.1 | 36.0 | 36.0 | 36.5 | 34.7 |

in the Split 4 of main paper. The performances of *FullyOracle* slightly vary across splits, because different numbers of novel classes are involved in the evaluation metric of different splits. Overall, in the scenario of more novel classes (*e.g.*, Split 9), the performances of our methods and baseline RETAB [14] are all degraded to some extent, due to the reduction of total annotations. Nevertheless, our method shows robustness even in the challenging Split 9, where the split ratio between base classes and novel classes is $1 : 1$. Therefore, the similarity in our method is highly and robustly transferrable across classes. Our method has preferable potential to learn more novel classes in practical and wide applications.