# OpenReview forum: "Weak-shot Semantic Segmentation via Dual Similarity Transfer"
_NeurIPS.cc/2022/Conference — NeurIPS 2022 Accept_

### Official Review · Reviewer_2A6K · 2022-07-09

**Rating:** 6
**Confidence:** 3
**Soundness:** 3 good
**Presentation:** 2 fair
**Contribution:** 2 fair

**Summary:**

This paper proposes a dual similarity transfer framework based on MaskFormer for the Weak-shot Semantic Segmentation task. It consists of the proposal-pixel similarity transfer and the pixel-pixel similarity transfer. Experiments are performed on the COCO-Stuff and ADE20K datasets.

**Questions:**

See questions in the Weaknesses part. Although the idea seems interesting, the presentation of this paper makes it boring to read, and it is hard to tell the approach's merit and reasonableness.


**Limitations:**

Limitations are discussed in the paper.

**Strengths And Weaknesses:**


**Strengths**
1. The results look good, and evaluations are extensive.
2. Considering the use of similarity transfer for segmentation makes sense to me.

**Weaknesses**
1. The presentation of this paper is poor.
- In view of the contribution of this work, the merit of the proposed work is unclear. For instance, in Line 77, contribution 1: "...(dual similarity transfer framework) is effective for proposal-pixel similarity transfer". It is meaningless since the "proposal-pixel similarity transfer" is one of the components of the dual similarity transfer framework.

- The method part is also hard to follow. It seems that this paper attempt to put the approach under the concept of similarity transfer [4]. However, the presentation of the method part mixes too many unrelated implementation details with the explanations of the design, making it hard to understand the reasonableness of the method. Based on my understanding, the proposed framework uses Maskformer as the base segmentation network, where the mask loss for novel classes is removed to fit the problem setting. It is a reasonable baseline, but I don't see a clear relation with the idea of "similarity transfer." Could there be more explanation about why it is called proposal-pixel similarity transfer?
 Besides, symbols are not reflected in Figure 2, making it hard to correspond with the text description. And there are too many natural language descriptions in Section 3.4. I think a formal mathematical description is needed to help understand how the method works.
  In addition, it seems that the proposed approach explicitly utilizes the "ignore labels" for learning masks of novel categories in Pixel-Pixel Similarity and Complementary Loss. I wonder whether it would cause the mask information leakage of the novel classes? If the novel and base classes do not appear in the same image, would the Complementary Loss fail to work? From another view, the results are close to the FullyOracle result on some split. Is it because of the possible information leakage?

- Quality of the figures is poor. Texts with light color in Figure 1(b) can not be seen clearly, and all figures would be blurred when zooming in.

---
EDIT
Although this work is somewhat incremental, I believe it is a meaningful attempt in WSSS with moderate-to-high impact in this sub-area. Therefore, I have increased my score from 4 to 6. The presentation may be further improved to make it easier to understand.

---

> ### Author Response · Authors · 2022-08-02
> **Respond to Reviewer 2A6K**
>
> __Q1__. In view of the contribution of this work, the merit of the proposed work is unclear. For instance, in Line 77, contribution 1: "...(dual similarity transfer framework) is effective for proposal-pixel similarity transfer". It is meaningless since the "proposal-pixel similarity transfer" is one of the components of the dual similarity transfer framework.
>
> __A1__. We choose to build our framework on MaskFormer, and MaskFormer lays the foundation for proposal-pixel similarity transfer. As introduced in Line 52-55, MaskFormer disentangles segmentation task into two sub-tasks: proposal classification and proposal segmentation, in which proposal segmentation corresponds to proposal-pixel similarity. We transfer such similarity from base classes to novel classes to obtain the high-quality segmentation results of novel classes, as shown in Tab. 2 and Fig. 4. Furthermore, our framework built on MaskFormer is more elegant, compared with the complex pipeline of RETAB [43] and WSSS methods [33,36] built on CAM [41], as discussed in Line 103-107. We have revised the summary of contribution 1 to make it more easily understood.
>
> ---
>
> __Q2__. I don't see a clear relation with the idea of "similarity transfer." Could there be more explanation about why it is called proposal-pixel similarity transfer?
>
> __A2__. As for “similarity”, the binary mask is produced by dot-product-sigmoid (See Line 132-134) between proposal embedding and pixel embeddings. Each entry in the binary mask indicates whether the proposal embedding is similar to each pixel embedding. As for “transfer”, proposal-pixel similarity is learned on base classes using GT masks of base classes and transferrable to novel classes. With novel proposal embeddings and proposal-pixel similarity, we can obtain the binary masks for novel classes by dot-product-sigmoid. Such similarity is pairwise semantic similarity and thus transferrable across different categories, which has also been exploited before [6,29,4]. Our experiments (Tab. 2,3) also provide in-depth support for the transferability.
>
> ---
>
> __Q3__. Symbols are not reflected in Figure 2, making it hard to correspond with the text description. And there are too many natural language descriptions in Section 3.3. I think a formal mathematical description is needed to help understand how the method works.
>
> __A3__. Thanks for your advice. We have modified Fig. 2 with symbols and improved the mathematical description in Section 3.3. We also have carefully revised the paper to make it more easily understood.
>
> ---
>
> __Q4__. In addition, it seems that the proposed approach explicitly utilizes the "ignore labels" for learning masks of novel categories in Pixel-Pixel Similarity and Complementary Loss. I wonder whether it would cause the mask information leakage of the novel classes? If the novel and base classes do not appear in the same image, would the Complementary Loss fail to work? From another view, the results are close to the FullyOracle result on some split. Is it because of the possible information leakage?
>
> __A4__. There is no information leakage of the novel classes, because all the not-base pixels (including both novel and ignore) are labeled as class 255 in implementation as stated in Line 221-223, so the model cannot see the real ignore labels.
>
> In the extreme cases, the Complementary Loss is still effective. If one image has only novel classes, the loss encourages the model to either predict high score for novel class or predict score lower than the ignore prior value. If one image has only base classes, the loss forces the model to either predict low score for novel class or predict score lower than the ignore prior value. The Complementary Loss provides valid information in both cases.
>
> Actually, the difficulty of learning novel classes depends not only on the novel classes themselves but also depends on the transferability from base classes to novel classes. Therefore, some splits may happen to fall into easy cases, and thus we use more than one random split to verify the effectiveness of our method.
>
> ---
>
> __Q5__. Quality of the figures is poor. Texts with light color in Figure 1(b) can not be seen clearly, and all figures would be blurred when zooming in.
>
> __A5__. Thanks for the suggestion. In Fig. 1 (b), considering that two images have duplicate ouputs, we use deeper color for mentioned variables while lighter color for unmentioned variables. We have revised Fig. 1 (b) and other figures for better quality.

---

> > ### Comment · Reviewer_2A6K · 2022-08-06
> > **Reviewer Response**
> >
> > Thank the authors for the response.
> > The rebuttal resolves most of my concerns about the contributions. Considering the contributions of this work, although somewhat incremental, I believe it is meaningful in WSSS.
> > The authors have revised the manuscript: 1. clarified the contributions (Q1), 2. provided explanations about how Maksformer correlates with similarity transfer (Q2), 3. made the illustrations in Fig. 2 clearer (Q3), 4. improved the quality of the figures (Q5).
> > Therefore, I would like to increase my score from 4 to 6.

---

> > > ### Author Response · Authors · 2022-08-07
> > > **Respond to Reviewer 2A6K**
> > >
> > > Thanks for your encouragement and recognizing our contribution!

---

### Official Review · Reviewer_B4BL · 2022-07-09

**Rating:** 6
**Confidence:** 4
**Ethics Flag:** Yes
**Soundness:** 3 good
**Presentation:** 3 good
**Contribution:** 3 good

**Summary:**

This paper focuses on the weak-shot semantic segmentation problem which consists of two groups of categories, i.e., one with both annotated categories and masks, while another one with only labeled categories.  For predicting precise segmentation masks, this paper proposes a dual similarity transfer network based on MaskFormer. Experiments are conducted on the COCO-Stuff-10K and ADE20K datasets to show the effectiveness.

**Questions:**

1. Line 162 - 164, 'Under ... produce a binary similarity with all the pixel embeddings.'.   Please explain how the similarities are learned and related to the supervision.
2. Is the SimNet trained in an end-to-end manner?
2. Line 187, what are the values of R_n ?
3. In Section 3.2, the notations should be specifically explained, e.g., i, w,h,c


**Limitations:**

Yes, limitations are adequately addressed.

**Strengths And Weaknesses:**

Strengths:
+ This paper is well organized and easy to follow.
+ This paper proposes to use a small network, i.e.,SimNet, to learn the pixel correlations between pixels within or across images with mask annotations, and further, SimNet is applied to estimate the pixel relations of images without mask annotations. Although this design introduces extra computations, this is effective in addressing the novel classes.
+ The experiments show that the proposed method can successfully improve the segmentation accuracy.

Weaknesses:
- According to the details in Section 4.2, pseudo-labels of novel classes generated by CAM are produced in advance. These pseudo-labels are utilized in the training process. However, the authors do not mention this in the method section. I think this part is crucial and should be included.

---

> ### Author Response · Authors · 2022-08-02
> **Respond to Reviewer B4BL**
>
> __Q1__. According to the details in Section 4.2, pseudo-labels of novel classes generated by CAM are produced in advance. These pseudo-labels are utilized in the training process. However, the authors do not mention this in the method section. I think this part is crucial and should be included.
>
> __A1__. As stated in Line 237-239, the re-trained version of our method uses the pseudo-labels of novel classes predicted by our model, instead of using CAM. As introduced in Line 233-236, the pseudo-labels generated by CAM is used for WSSS baselines and RETAB for training the segmentation model.  As shown in Tab 1, even without re-training, our method can achieve satisfying performance and could be further improved after the commonly used re-training.
>
> ---
>
> __Q2__. Line 162 - 164, 'Under ... produce a binary similarity with all the pixel embeddings.'. Please explain how the similarities are learned and related to the supervision.
>
> __A2__. Specifically, we compute the dot-production-sigmoid between each proposal embedding and each pixel embedding, where the dot-production corresponds to a similarity function. For each base class, we have its GT mask indicating whether the pixel belongs to the base class, and thus we have the supervision on the binary similarity between base proposal embedding and pixel embeddings. Specifically, the supervision is the mask loss in Eqn. 2 for similarity learning.
>
> ---
>
> __Q3__. Is the SimNet trained in an end-to-end manner?
>
> __A3__. Yes, the whole framework in Fig. 2 is trained end-to-end (one-stage).
>
> ---
>
> __Q4__. Line 189, what are the values of R_n ?
>
> __A4__. The value of R_n is the pixel-pixel similarity estimated by the SimNet, indicating whether the input pixel pair comes from the same class.
>
> ---
>
> __Q5__. In Section 3.2, the notations should be specifically explained, e.g., i, w,h,c.
>
> __A5__. Thanks for the advice. Specifically, [H,W] is the spatial size of pixel embeddings and C is the channel number of feature. We use [h,w] to indicate the spatial index of pixel and c to indicate the class index in K classes totally.

---

> > ### Comment · Reviewer_B4BL · 2022-08-08
> > **Reviewer Respond**
> >
> > Thanks for the response.
> >
> > The authors have replied my question and resolved my concerns.  I am happy to recommend this paper to be accepted.
> >
> > In addition, the proposed method have  multiple modules and steps, it is suggested the authors can release the source and trained model if it get accepted finally, which would be easier for future researchers to reproduce.

---

> > > ### Author Response · Authors · 2022-08-08
> > > **Respond to Reviewer B4BL**
> > >
> > > Thanks for your recommendation. We are pleased to release the code and model for future researchers to further explore.

---

### Official Review · Reviewer_WaRz · 2022-07-20

**Rating:** 4
**Confidence:** 1
**Ethics Flag:** Yes
**Soundness:** 2 fair
**Presentation:** 1 poor
**Contribution:** 3 good

**Summary:**

This paper proposes a novel method on weak-shot semantic segmentation, in which fully-supervised model, MaskFormer, with dual similarity transfer was employed. The basic idea is totaly diffrent from the existing method of weak-shot segmentation, RETAB, which is based on the WSSS method. The experimental results showed the effectiveness of the proposed method.

**Questions:**

1) Why sampling only 100 pixels (J=100) is enough ? It seems to be very small compared to the total pixels of an image.
2) \gamma was examined in the ablation studies. Where \gamma was used? Eq.5 does not contain \gamma.
3) The reviewer is not sure why the binary mask estimation model trained with base classes are applicable for novel class.
4) It is unclear how to train the model using base classes which have pixel-wise annotation and novel classes which have only image-level labels.

**Limitations:**

In Sup. Sec.8, the limitation is writen concretely. Since the limitation discussion is important, part of it should be included in the mail text.

**Strengths And Weaknesses:**

Pros)
 + New approarch for weak-shot segmentation task
 + The results by the proposed method outperformed the baseline, RETAB.

Cons)
 + The paper is very hard to understand. It is unclear why the proposed method works well.
 + Only one baseline, RETAB, was used. No meaning to compare with WSSS in which no pixel-wise annotation was used at all.

---

> ### Author Response · Authors · 2022-08-02
> **Respond to Reviewer WaRz**
>
> __Q1__. The paper is very hard to understand. It is unclear why the proposed method works well.
>
> __A1__. We have carefully revised the paper to make it more easily understood.
>
> We build our method on MaskFormer, which is relatively harder to understand compared with typical segmentation model (e.g., FCN). The key insight of MaskFormer is to disentangle the semantic segmentation task into two sub-tasks: proposal classification and proposal segmentation, as introduced in Line 52-56. The first sub-task assigns class labels to proposal embeddings. The second sub-task calculates the similarity between each proposal embedding and all pixels to obtain a similarity map, which is the segmentation mask corresponding to the class of this proposal embedding.
>
> Based on the two sub-tasks of MaskFormer, our proposed method could be easily understood. For proposal classification, base classes and novel classes both have image-level labels for learning. For proposal segmentation, novel classes depend on the dual similarity transferred from base classes for learning. See A2/A3 below for further explanation.
>
> ---
>
> __Q2__. The reviewer is not sure why the binary mask estimation model trained with base classes is applicable for novel class.
>
> __A2__. Following A1, it is applicable because the similarity (binary mask) is learned on base classes and could be transferred to novel classes. Specifically, base proposal embeddings and novel proposal embeddings are learned using image-level labels. The proposal-pixel similarity (dot-product-sigmoid) is learned using pixel-level labels of base classes. With novel proposal embeddings and proposal-pixel similarity, we can obtain the binary masks for novel classes by the same dot-product-sigmoid. The similarity is pairwise semantic similarity and thus transferrable across different categories, which has also been exploited before [6,29,4]. Our experiments (Tab. 2,3) also provide in-depth support for the transferability.
>
> ---
>
> __Q3__. It is unclear how to train the model using base classes which have pixel-wise annotation and novel classes which have only image-level labels.
>
> __A3__. The key is that the two sub-tasks for segmentation are disentangled. We can supervise two sub-tasks for base classes and only one sub-task for novel classes. The different supervisions are applied according to the assigned classes respectively (see the class selection in equations and loss color in Fig. 2). For example, the mask loss is only applied to the binary masks of base classes and the classification loss is applied to both base classes and novel classes.
>
> ---
>
> __Q4__. Only one baseline, RETAB, was used. No meaning to compare with WSSS in which no pixel-wise annotation was used at all.
>
> __A4__. The WSSS baselines we compared are not standard WSSS methods. As described in Line 231-235, we use pixel-wise annotations of base classes in the re-training stage. In other words, these WSSS methods use the same supervision as our method for a fair comparison.
>
> ---
>
> __Q5__. Why sampling only 100 pixels (J=100) is enough ? It seems to be very small compared to the total pixels of an image.
>
> __A5__. On the one hand, the feature map has a relatively lower resolution (160x160) . On the other hand, the number of pixel pairs is 100x100, which is relatively enough for each training iteration. The ablation study in Tab. 1 in Supplementary indicates that using more pixels (e.g., 150) only slightly improves the performance. Our default value is a reasonable choice considering the computation consumption.
>
> ---
>
> __Q6__. \gamma was examined in the ablation studies. Where \gamma was used? Eq.5 does not contain \gamma.
>
> __A6__. As stated in Line 278-279, \gamma represents the prior probability for the ignore class, similar to the background threshold used in weakly supervised semantic segmentation [17,33,35]. More detailed descriptions can be found in Line 207-210.

---

### Meta-Review · Area_Chair_UMAg · 2022-08-24

**Recommendation:** Accept
**Confidence:** Certain

**Metareview:**

Two reviewers give a weak accept rating while the other one gives a borderline reject rating. Considering the low confidence of the negative comment and the contrary comments in paper writing (confident "easy to follow" vs. unconfident "hard to understand"), the AC would lean to accept this paper.

**Award:**

No

---

### Decision · Program_Chairs · 2022-09-14

Accept